# Stochastic Stabilization of Dual-Layer Rumor Propagation Model with Multiple Channels and Rumor-Detection Mechanism

**DOI:** 10.3390/e25081192

**Published:** 2023-08-10

**Authors:** Xiaojing Zhong, Chaolong Luo, Xiaowu Dong, Dingyong Bai, Guiyun Liu, Ying Xie, Yuqing Peng

**Affiliations:** 1School of Mechanical and Electrical Engineering, Guangzhou University, Guangzhou 510006, China; zhongxj@gzhu.edu.cn (X.Z.); 2112207035@e.gzhu.edu.cn (C.L.); dong_xiaowu@163.com (X.D.); liugy@gzhu.edu.cn (G.L.); 2School of Mathematics, Guangzhou University, Guangzhou 510006, China; baidy@gzhu.edu.cn; 3School of Public Administraion, Guangzhou University, Guangzhou 510006, China; 4School of Journalism and Communication, Guangzhou University, Guangzhou 510006, China

**Keywords:** stochastic stabilization, dual-layer rumor propagation model, rumor detection mechanism

## Abstract

With the development of information technology, individuals are able to receive rumor information through various channels and subsequently act based on their own perceptions. The significance of the disparity between media and individual cognition in the propagation of rumors cannot be underestimated. In this paper, we establish a dual-layer rumor propagation model considering the differences in individual cognition to study the propagation behavior of rumors in multiple channels. Firstly, we obtain the threshold for rumor disappearance or persistence by solving the equilibrium points and their stability. The threshold is related to the number of media outlets and the number of rumor debunkers. Moreover, we have innovatively designed a class of non-periodic intermittent noise stabilization methods to suppress rumor propagation. This method can effectively control rumor propagation based on a flexible control scheme, and we provide specific expressions for the control intensity. Finally, we have validated the accuracy of the theoretical proofs through experimental simulations.

## 1. Introduction

With the popularity of the Internet and the rise of social media, the speed and scope of information dissemination have greatly expanded. However, this has also provided a wider and more efficient channel for rumor propagation. On the Internet, users can forward, comment on, and analyze unverified information through social software, as well as obtain rumor information through websites, allowing rumors to spread rapidly to millions or even billions of users in a short period of time [1]. As a result, the spread and impact of rumors far exceed those of the traditional media era. The rapid spread of rumors makes it easy for them to trigger public panic and confusion in a short period of time, which has adverse effects on social order and stability. Therefore, it is crucial to study the propagation trends of rumors in different media environments and design effective rumor-control strategies.

In the analysis of rumor propagation research, the main approach is to establish mathematical models to study the dynamic behavior of rumor propagation, drawing to some extent from models used in the study of disease transmission. The classical rumor propagation model is the *DK* rumor propagation model proposed by Daley and Kendall based on infectious disease transmission models [2], and Maki and Thomson optimized the *DK* model to propose the *MK* model [3]. In order to better describe the process of rumor propagation, researchers have established many valuable new rumor propagation models by considering factors such as individual behavioral differences and environmental influences [3,4,5,6,7,8,9,10,11,12,13]. In terms of the propagation environment, researchers [3,4] have considered the influence of random factors in the propagation environment and established new stochastic rumor propagation models. Li Jiarong et al. [5] considered the mixture of multiple languages in the propagation environment and established a class of ${IS}_{1}S_{2}R_{1}R_{2}$ propagation models. In terms of individual behavioral differences, researchers have established various rumor propagation models with their own characteristics from different perspectives such as initial states [6], opinion leaders [7], repeated forwarding behavior [8], biased absorption behavior [9], and individual cognitive level [10,11]. We would like to highlight that a new type of C node, characterizing true verification individuals (checkers), has been proposed in reference [14]. The researchers’ newly established ISSC model can reflect the effects of fake news to avoid or attenuate them. In addition, Cui et al. [12] consider that individuals like to concurrently use multiple social networks and show different passions for information acceptance, which they called individual fashion-passion trend (IFPT) characteristics. Then, they establish a two-layer rumor propagation model. Liu et al. [13] analyzed the propagation behavior of information in the dual-subject mode of mobile devices and crowds and established a class of *URBD* user-devices two-layer models. Huo et al. [15] constructed an improved *XYZ-ISR* two-layer model describing the dynamic process of rumor propagation in multiple channels. Based on the above models, researchers have used methods such as next-generation matrices and Lyapunov functions to explore the propagation dynamics properties such as propagation thresholds and equilibrium point stability. However, a comprehensive two-layer model that considers both media and individual cognitive levels does not currently exist, and the influence of both on rumor propagation still requires further in-depth research.

In the field of rumor control research, the current main control strategies are primarily deterministic control approaches. These include direct blocking of rumor spreaders, such as achieving account suspension, speech restriction [16,17], and cutting off propagation paths through algorithm adjustments [18]; or dissemination channels [19,20,21,22,23], disclosure of rumor information by media or individuals [24,25,26,27,28,29], and so on. Zhu et al. [16] proposed a control strategy called Unified Immune Control, which involves proportionally blocking information spreaders. Manouchehri et al. [19] introduced the idea of employing verified states for verification purposes in combating fake news. Zhu et al. [23] explored the use of quarantine measures and developed a *SIRQU* model to assess the control efficiency and feasibility of blocking rumor propagation. Li, et al. [30] designed a debunking mechanism targeting misinformation and studied the effectiveness of debunking strategies in guiding the direction of propagation from both a global and local optimization perspective. Huo et al. [31] extensively considered the positive guiding role of the media and established a dual-network propagation system that coupled media and nodes. They combined event-driven pulse control schemes to control the direction of information propagation. Many researchers integrated these two control strategies to design hybrid mixed-control strategies, which involved the design of specific control strategies from various continuous, discrete, and other perspectives, and verified the effectiveness of these control strategies. It should be noted that deterministic control strategies mentioned above face challenges during implementation, such as difficulties in identifying hidden propagators and accurately locating control targets, as well as complex network connectivity that may violate security boundaries. In contrast to deterministic control strategies, stochastic stabilization strategies do not fix control targets and offer relatively more flexibility in the control approach.

The stochastic stabilization method described by white noise has been proven effective in both theoretical and applied contexts. The theoretical research on stochastic stabilization described by white noise was proposed and demonstrated by Xuerong Mao in the framework of Ito calculus in 2008 [24]. In recent years, discrete-time stochastic control methods have gained increasing attention, including discrete feedback control [25,26,27] and intermittent control [28,29,32]. It is worth noting that the research on stochastic intermittent noise control methods is just emerging, and further exploration is needed to uncover its significant role in suppressing rumors.

Based on the above analysis, this paper proposes a *XYZ-ISTD* dual-layer rumor propagation model considering the media communication rumor-detection mechanism. The contributions of this paper are as follows:Considering the individual differences in rumor detection and the influence of media in rumor propagation, an *XYZ-ISTD* dual-layer rumor propagation model is proposed, which explores the effects of debunking and rumor propagation.The paper theoretically analyzes how the counter-rumor mechanism lowers the threshold of rumor existence, thereby exerting a positive effect on suppressing rumor propagation.A non-periodic intermittent control strategy is designed to suppress rumor propagation, which allows one to control the rumor propagation system according to ideal requirements while considering the control cost.

The rest of the paper is organised as follows. In Section 2, a dual-layer *XYZ-ISTD* network propagation model with rumor detection mechanism is developed. In Section 3, the stability analysis of the propagation model is given. In Section 4, a non-periodic stochastic intermittent control strategy is designed and the stability of the controlled system is analyzed. In Section 5, we demonstrate the validity of the proposed theory through numerical simulations. In Section 6, we present our conclusions.

## 2. Model Formulation

To better portray real communication scenarios, the role of the media and rumor screeners in influencing rumor propagation is examined. We divide the model structure into the media layer (XYZ) and the population layer (ISTD). The media layer consists of susceptible message media X(t), representing information media that do not propagate rumors; infected message media Y(t), representing information media that propagate rumors; and immune message media Z(t), representing information media that propagate both rumors and truths. The population layer consists of ignoramuses I(t), representing individuals who are unaware of the existence of rumors but are susceptible to their influence; spreaders S(t), representing individuals who, upon exposure to rumors, believe and propagate them; stiflers T(t), representing individuals who believe in rumors and are forcibly isolated, no longer participating in rumor propagation; and debunkers D(t), representing individuals who believe in debunking rumors and propagating truths to expose rumors. For simplicity, we name the rumor propagation model as the XYZ−ISDT model. The rumor propagation mechanism of the XYZ−ISDT model can be described as follows:(1)We assume that rumors spread within a reasonable period of time, with a constant influx of individuals denoted by *A* per unit time, and each category has an outflow of individuals denoted by μ.(2)When ignoramuses come into contact with spreaders, their belief in rumors leads them to become spreaders with a probability of $\beta_{1}$. At the same time, ignoramuses may come into contact with debunkers and be willing to propagate truthful information with a probability of $\beta_{3}$, becoming debunkers.(3)When spreaders come into contact with debunkers, the influence of truths leads them to become debunkers with a probability of $\beta_{2}$.(4)To suppress rumor propagation, we isolate spreaders with a probability of η, making them stiflers.(5)Spreaders propagate rumors in the media, causing susceptible message media to become infected message media with a probability of $\alpha_{1}$. At the same time, infected message media further promote rumor propagation, making ignoramuses become spreaders with a probability of $\gamma_{1}$.(6)Debunkers propagate truths in the media to suppress rumors, causing infected message media to become immune message media with a probability of $\alpha_{2}$. At the same time, immune message media further suppress rumor propagation, making spreaders become debunkers with a probability of $\gamma_{2}$.

Based on the above dynamics, we construct the state transition diagram of rumor propagation as shown in Figure 1. The rumor propagation model of the dual-layer network is represented by the following equations:(1)$$\left\{\begin{array}{c} \frac{dX(t)}{dt}=-\alpha_{1}X\left(t\right)S\left(t\right),\\ \frac{dY(t)}{dt}=\alpha_{1}X\left(t\right)S\left(t\right)-\alpha_{2}Y\left(t\right)D\left(t\right),\\ \frac{dZ(t)}{dt}=\alpha_{2}Y\left(t\right)D\left(t\right),\\ \frac{dI(t)}{dt}=A-\beta_{1}I\left(t\right)S\left(t\right)-\gamma_{1}I\left(t\right)Y\left(t\right)-\beta_{3}I\left(t\right)D\left(t\right)-\mu I\left(t\right),\\ \frac{dS(t)}{dt}=\beta_{1}I\left(t\right)S\left(t\right)+\gamma_{1}I\left(t\right)Y\left(t\right)-\beta_{2}S\left(t\right)D\left(t\right)-\gamma_{2}S\left(t\right)Z\left(t\right)-\left(\mu +\eta \right)S\left(t\right),\\ \frac{dT(t)}{dt}=\eta S\left(t\right)-\mu T\left(t\right),\\ \frac{dD(t)}{dt}=\beta_{2}S\left(t\right)D\left(t\right)+\gamma_{2}S\left(t\right)Z\left(t\right)+\beta_{3}I\left(t\right)D\left(t\right)-\mu D\left(t\right), \end{array}\right.$$
where the initial values X(0)>0, Y(0)>0, Z(0)>0, I(0)>0, S(0)>0, T(0)>0, and D(0)>0. All of the parameter values are assumed to be nonnegative and *A*, μ > 0. The meanings of the parameters are given in Table 1.

Set $N_{1}\left(t\right)$ to be the size of the overall in the media layer at moment *t* and $N_{2}\left(t\right)$ to be the size of the overall individuals in the population layer at moment *t*, where $N_{1}\left(t\right)=X\left(t\right)+Y\left(t\right)+Z\left(t\right)$, $N_{2}\left(t\right)=I\left(t\right)+S\left(t\right)+T\left(t\right)+D\left(t\right)$.

Adding the fist three equations and the last four equations to system (Equation 1) gives
$$\begin{array}{c} \begin{array}{ccc} N_{1}^{{}^{\prime }}\left(t\right) & = & 0,\\ N_{2}^{{}^{\prime }}\left(t\right) & = & {\left(I(t)+S(t)+T(t)+D(t)\right)}^{{}^{\prime }}\\ & = & A-\mu \left(I(t)+S(t)+T(t)+D(t)\right)\\ & = & A-\mu N_{2}\left(t\right). \end{array} \end{array}$$
combining with the initial values that $$N_{1}\left(0\right)=N_{1},\;N_{2}\left(0\right)=I\left(0\right)+S\left(0\right)+T\left(0\right)+D\left(0\right)\leq \frac{A}{\mu },$$ we obtain
$$\underset{t\to \infty }{lim sup}N_{1}\left(t\right)=N_{1},\;\underset{t\to \infty }{lim sup}N_{2}\left(t\right)=\frac{A}{\mu }.$$

Therefore, all solutions are bounded and the feasible region for system (Equation 1) is
$$\Gamma =\left\{\begin{array}{c} \left(X(t),Y(t),Z(t),I(t),S(t),T(t),D(t)\right)\in R_{+}^{7n}|\\ X\left(t\right)+Y\left(t\right)+Z\left(t\right)=N_{1},I\left(t\right)+S\left(t\right)+T\left(t\right)+D\left(t\right)\leq \frac{A}{\mu } \end{array}\right\}$$

To guarantee the positivity of the solution of system (Equation 1), we conclude Theorem 1 to guantee it.

**Theorem 1.** 
*For any initial values X(0)>0,Y(0)>0,Z(0)>0,I(0)>0,S(0)>0,T(0)>0,D(0)>0, the solutions of system (1) are positive for all t≥0.*


**Proof.** For any initial values X(0)>0,Y(0)>0,Z(0)>0,I(0)>0,S(0)>0,T(0)>0,D(0)>0. In order to verify the positivity of the solution, we first proved the positivity of X(t)>0,Y(t)>0 and Z(t)>0.Integrating over $\frac{dX(t)}{dt}=-\alpha_{1}X\left(t\right)S\left(t\right)$, we can obtain $X\left(t\right)=X\left(0\right)e^{\int_{0}^{t}-\alpha_{1}S\left(\zeta \right)d\zeta }$. Since X(0)>0, we can obtain X(t)>0 for t>0. Integrating over $\frac{dY(t)}{dt}=\alpha_{1}X\left(t\right)S\left(t\right)-\alpha_{2}Y\left(t\right)D\left(t\right)$, we can obtain $Y\left(t\right)=e^{\int_{0}^{t}-\alpha_{2}D\left(\zeta \right)d\zeta }\left(\int_{0}^{t}\alpha_{1}X\left(\zeta_{1}\right)S\left(\zeta_{1}\right)e^{\int_{0}^{\zeta_{1}}\alpha_{2}D\left(\zeta_{2}\right)d\zeta_{2}}d\zeta_{1}+Y\left(0\right)\right)$. Since Y(0)>0, we can obtain Y(t)>0 for t>0. Similarly, the provable Z(t)>0 for t>0.We denote $\chi \left(t\right)=\min_{t}\left\{I(t),S(t),T(t),D(t)\right\}$. By using reduction to absurdity, we assume there exists a constant τ>0, such that
$$X\left(t\right)>0,t\in \left(0,\tau \right),and\;X\left(\tau \right)=0,\dot{X}\left(\tau \right)\leq 0.$$From the definition of X(t), X(τ)=0 implies I(τ)=0,S(τ)=0,T(τ)=0 or D(τ)=0.If χ(τ)=I(τ), the model (1) implies that
$$\frac{dI(\tau )}{dt}=A>0,$$
a contradiction to $\dot{\chi }\left(\tau \right)\leq 0$. Hence, I(t)>0.If χ(τ)=S(τ), the model (1) implies that
$$\frac{dS(\tau )}{dt}=\gamma_{1}I\left(\tau \right)Y\left(\tau \right)>0,$$
a contradiction to $\dot{\chi }\left(\tau \right)\leq 0$. Hence, S(t)>0.If χ(τ)=T(τ), the model (1) implies that
$$\frac{dT(\tau )}{dt}=\eta S\left(\tau \right)>0,$$
a contradiction to $\dot{\chi }\left(\tau \right)\leq 0$. Hence, T(t)>0.If χ(τ)=D(τ), the model (1) implies that
$$\frac{dD(\tau )}{dt}=\gamma_{2}S\left(t\right)Z\left(t\right)>0,$$
a contradiction to $\dot{\chi }\left(\tau \right)\leq 0$. Hence, D(t)>0. The proof is completed. □

## 3. Dynamic Analysis

In this section, we focus on the dynamic behavior of system (Equation 1). Based on this, we can obtain criteria for the disappearance and persistence of rumor spreaders and the positive inhibitory effect of the detection mechanism.

### 3.1. Existence of Equilibrium Points

In order to gain the threshold of whether a rumor naturally dies out or spreads, we need to demonstrate the existence and stability of the equilibrium points. First, we aim to establish the existence of equilibrium points.

**Definition 1.** 
*Define the equilibrium point*

E=(X(∞),Y(∞),Z(∞),I(∞),S(∞),T(∞),D(∞))

*for system (1). When S(∞)=0, we treat system (1) as if there is a rumor-free equilibrium $E_{f}$, and if system (1) reaches $E_{f}=\left\{E|S(\infty )=0\right\}$, then the system is rumor-free; when S(∞)>0, we treat system (1) as if there is a local rumor equilibrium $E_{e}$, and if system (1) reaches $E_{e}=\left\{E|S(\infty )>0\right\}$, then system (1) is rumor-prevalent.*


Letting the right side of system (Equation 1) equals zero, we obtain the equilibrium equation for system (Equation 1) as follows:(2)$$\left\{\begin{array}{c} -\alpha_{1}X\left(\infty \right)S\left(\infty \right)=0,\\ \alpha_{1}X\left(\infty \right)S\left(\infty \right)-\alpha_{2}Y\left(\infty \right)D\left(\infty \right)=0,\\ \alpha_{2}Y\left(\infty \right)D\left(\infty \right)=0,\\ A-\beta_{1}I\left(\infty \right)S\left(\infty \right)-\gamma_{1}I\left(\infty \right)Y\left(\infty \right)-\mu I\left(\infty \right)-\beta_{3}I\left(\infty \right)D\left(\infty \right)=0,\\ \beta_{1}I\left(\infty \right)S\left(\infty \right)+\gamma_{1}I\left(\infty \right)Y\left(\infty \right)-\beta_{2}S\left(\infty \right)D\left(\infty \right)-\left(\mu +\eta \right)S\left(\infty \right)-\gamma_{2}S\left(\infty \right)Z\left(\infty \right)=0,\\ \eta S(\infty )-\mu T(\infty )=0,\\ \beta_{2}S\left(\infty \right)D\left(\infty \right)+\gamma_{2}S\left(\infty \right)Z\left(\infty \right)+\beta_{3}I\left(\infty \right)D\left(\infty \right)-\mu D\left(\infty \right)=0. \end{array}\right.$$

For the case of no rumor, let S(∞)=0; then, $E_{f}$ consists of two types of equilibrium $E_{f1}=\left(0,0,N_{1},\frac{A}{\mu },0,0,0\right)$ and $E_{f2}=\left(0,0,N_{1},\frac{\mu }{\beta_{3}},0,0,\frac{A\beta_{3}-\mu^{2}}{\beta_{3}\mu }\right)$, where the conditions for the existence of $E_{f2}$ are
$$\frac{\mu }{\beta_{3}}<\frac{A}{\mu },\frac{A\beta_{3}-\mu^{2}}{\beta_{3}\mu }<\frac{A}{\mu }.$$
That is, $A\beta_{3}-\mu^{2}>0$. For the case of the prevalent rumor, solving system (Equation 2) gives $E_{e}=\left(0,0,N_{1},I^{*},S^{*},T^{*},D^{*}\right)$.

Next, we discuss the stability behavior of the rumor-free equilibrium point $E_{f}$ of system (Equation 1).

### 3.2. Stability Analysis of the Rumor-Free Equilibrium Point $E_{f1}$


**Theorem 2.** 
*If $\max\left(\frac{\beta_{1}A}{\mu \left(\mu +\eta +\gamma_{2}N_{1}\right)},\frac{\beta_{3}A}{\mu^{2}}\right)<1$, the rumor-free equilibrium point $E_{f1}$ of system (1) is local asymptotically stable.*


**Proof.** The Jacobian matrix $J(E_{f1})$ of system (Equation 1) at the rumor-free equilibrium point $E_{f1}$ is
$$J\left(E_{f1}\right)=\left[\begin{array}{ccccccc} 0 & 0 & 0 & 0 & 0 & 0 & 0\\ 0 & 0 & 0 & 0 & 0 & 0 & 0\\ 0 & 0 & 0 & 0 & 0 & 0 & 0\\ 0 & -\gamma_{1}\frac{A}{\mu } & 0 & -\mu & -\beta_{1}\frac{A}{\mu } & 0 & -\beta_{3}\frac{A}{\mu }\\ 0 & \gamma_{1}\frac{A}{\mu } & 0 & 0 & \beta_{1}\frac{A}{\mu }-\mu -\eta -\gamma_{2}N_{1} & 0 & 0\\ 0 & 0 & 0 & 0 & \eta & -\mu & 0\\ 0 & 0 & 0 & 0 & \gamma_{2}N_{1} & 0 & \beta_{3}\frac{A}{\mu }-\mu \end{array}\right].$$According to $\left|\lambda E-J(E_{f1})\right|=0$, we obtain the characteristic equation
(3)$$\lambda^{3}{\left(\lambda +\mu \right)}^{2}\left(\lambda -\beta_{1}\frac{A}{\mu }+\mu +\eta +\gamma_{2}N_{1}\right)\left(\lambda -\beta_{3}\frac{A}{\mu }+\mu \right)=0.$$
Obviously, it is easy to obtain the four eigenvalues as follows: $\lambda_{1,2,3}=0$, $\lambda_{4,5}=-\mu$, which are all non-positive. As for the sixth and seventh eigenvalue of Equation (Equation 3), they satisfy the following equations:
$$\lambda_{6}=\beta_{1}\frac{A}{\mu }-\left(\mu +\eta +\gamma_{2}N_{1}\right)=\left(\mu +\eta +\gamma_{2}N_{1}\right)\left(\frac{\beta_{1}A}{\mu \left(\mu +\eta +\gamma_{2}N_{1}\right)}-1\right),$$
$$\lambda_{7}=\beta_{3}\frac{A}{\mu }-\mu =\mu \left(\frac{\beta_{3}A}{\mu^{2}}-1\right).$$So $\max\left(\frac{\beta_{1}A}{\mu \left(\mu +\eta +\gamma_{2}N_{1}\right)},\frac{\beta_{3}A}{\mu^{2}}\right)<1$; then, the rumor-free equilibrium point $E_{f1}$ is locally asymptotically stable. □

### 3.3. Stability Analysis of the Rumor-Free Equilibrium Point $E_{f2}$


**Theorem 3.** 
*If $\frac{\beta_{1}\mu^{2}-\beta_{2}\left(A\beta_{3}-\mu^{2}\right)}{\beta_{3}\mu \left(\mu +\eta +\gamma_{2}N_{1}\right)}<1$ and $A\beta_{3}-\mu^{2}>0$, the rumor-free equilibrium point $E_{f2}$ of system (Equation 1) is locally asymptotically stable.*


**Proof.** The Jacobian matrix $J(E_{f2})$ of system (Equation 1) at the rumor-free equilibrium point $E_{f2}$ is
$$J\left(E_{f2}\right)=\left[\begin{array}{ccccccc} 0 & 0 & 0 & 0 & 0 & 0 & 0\\ 0 & -\alpha_{2}\rho & 0 & 0 & 0 & 0 & 0\\ 0 & \alpha_{2}\rho & 0 & 0 & 0 & 0 & 0\\ 0 & -\gamma_{1}\frac{\mu }{\beta_{3}} & 0 & -\mu -\rho & -\beta_{1}\frac{\mu }{\beta_{3}} & 0 & -\mu \\ 0 & \gamma_{1}\frac{\mu }{\beta_{3}} & 0 & 0 & \beta_{1}\frac{\mu }{\beta_{3}}-\beta_{2}\rho -\mu -\eta -\gamma_{2}N_{1} & 0 & 0\\ 0 & 0 & 0 & 0 & \eta & -\mu & 0\\ 0 & 0 & 0 & \rho & \beta_{2}\rho +\gamma_{2}N_{1} & 0 & 0 \end{array}\right],$$
where $\rho =\frac{A\beta_{3}-\mu^{2}}{\beta_{3}\mu }$.According to $\left|\lambda E-J(E_{f2})\right|=0$, we obtain the characteristic equation
(4)$$\begin{array}{c} \lambda^{2}\left(\lambda +\mu \right)\left(\lambda +\alpha_{2}\frac{A\beta_{3}-\mu^{2}}{\beta_{3}\mu }\right)\left(\lambda^{2}+\lambda \mu +\lambda \frac{A\beta_{3}-\mu^{2}}{\mu }+A\beta_{3}-\mu^{2}\right)\\ (\lambda -\beta_{1}\frac{\mu }{\beta_{3}}+\beta_{2}\frac{A\beta_{3}-\mu^{2}}{\beta_{3}\mu }+\mu +\eta +\gamma_{2}N_{1})=0. \end{array}$$
Obviously, it is easy to obtain the six eigenvalues as follows:
$$\begin{array}{ccc} \lambda_{1,2} & = & 0,\\ \lambda_{3} & = & -\mu ,\\ \lambda_{4} & = & -\alpha_{2}\frac{A\beta_{3}-\mu^{2}}{\beta_{3}\mu },\\ \lambda_{5} & = & \frac{-\left(\mu +\frac{A\beta_{3}-\mu^{2}}{\mu }\right)-\sqrt{{\left(\mu +\frac{A\beta_{3}-\mu^{2}}{\mu }\right)}^{2}-4\left(A\beta_{3}-\mu^{2}\right)}}{2},\\ \lambda_{6} & = & \frac{-\left(\mu +\frac{A\beta_{3}-\mu^{2}}{\mu }\right)+\sqrt{{\left(\mu +\frac{A\beta_{3}-\mu^{2}}{\mu }\right)}^{2}-4\left(A\beta_{3}-\mu^{2}\right)}}{2}, \end{array}$$
which are all non-positive. As for the seventh eigenvalue of Equation (Equation 4), which satisfies the following equation:
$$\xi_{7}=\beta_{1}\frac{\mu }{\beta_{3}}-\beta_{2}\frac{A\beta_{3}-\mu^{2}}{\beta_{3}\mu }-\left(\mu +\eta +\gamma_{2}N_{1}\right)=\left(\mu +\eta +\gamma_{2}N_{1}\right)\left(\frac{\beta_{1}\mu^{2}-\beta_{2}\left(A\beta_{3}-\mu^{2}\right)}{\beta_{3}\mu \left(\mu +\eta +\gamma_{2}N_{1}\right)}-1\right).$$So, $\frac{\beta_{1}\mu^{2}-\beta_{2}\left(A\beta_{3}-\mu^{2}\right)}{\beta_{3}\mu \left(\mu +\eta +\gamma_{2}N_{1}\right)}$ and $A\beta_{3}-\mu^{2}>0$; then, the rumor-free equilibrium point $E_{f2}$ is local asymptotically stable. □

**Remark 1.** 
*The conditions of Theorem 2 can be expressed in the form $R_{0}=\max\left(\frac{\beta_{1}A}{\mu \left(\mu +\eta +\gamma_{2}N_{1}\right)},\frac{\beta_{3}A}{\mu^{2}}\right)<1$, which represents the typical propagation threshold [3,4,5,6,7,8,13] without the detection mechanism. Observing the conditions of Theorem 3, $A\beta_{3}-\mu^{2}>0$ deduces $\frac{A\beta_{3}}{\mu^{2}}>1$, which means $R_{0}>1$. Moreover, $A\beta_{3}-\mu^{2}>0$ deduces $\frac{A}{\mu }-\frac{\mu }{\beta_{3}}>0$; then,*

$$\frac{\beta_{1}A}{\mu \left(\mu +\eta +\gamma_{2}N_{1}\right)}>\frac{\beta_{1}\mu }{\beta_{3}\left(\mu +\eta +\gamma_{2}N_{1}\right)},$$

*which means*

$$\frac{\beta_{1}A}{\mu \left(\mu +\eta +\gamma_{2}N_{1}\right)}>\frac{\beta_{1}\mu }{\beta_{3}\left(\mu +\eta +\gamma_{2}N_{1}\right)}-\frac{\beta_{2}\left(A\beta_{3}-\mu^{2}\right)}{\beta_{3}\mu \left(\mu +\eta +\gamma_{2}N_{1}\right)}.$$


*Namely, if $R_{0}>1$, according to the conclusions presented in [13], the rumor will spread. However, Theorem 3 proved that the rumor will not widely propagate even $R_{0}>1$. This effectively demonstrates the inhibitory effect of the rumor detection mechanism on rumor propagation.*


## 4. Non-Periodically Intermittent Stochastic Stabilization Strategy

The calming effect of white noise can create a relatively calm and objective environment, which helps one to approach rumors more rationally. Inspired by this, this section proposes a non-periodic random control strategy for rumor propagation. We choose Brownian motion to describe the random selection process, and the controller is composed of Gaussian white noise. The stochastic stabilization system can be represented as follows:dx=f(x,t)dt−g(x,t)dB(t).

In particular, we consider the efficiency of cost control and the flexibility of control time by dividing each time interval $\left[t_{k},t_{k+1}\right]$ into control working time $\left[t_{k},t_{k}+\omega_{k}\right)$ and control resting time $\left[t_{k}+\omega_{k},t_{k+1}\right)$, where $\omega_{k}$ represents the control width for the *k*th interval. The starting time $t_{k}$ and control width $\omega_{k}$ vary, and the total control time satisfies the following inequality compared to *w*:$$\lim_{n\to \infty }\frac{\sum_{j+1}^{n}\omega_{j}}{t_{n+1}-t_{0}}=\omega .$$

Taking into account the differences between the media and the general population, we introduce different stochastic controllers for the media layer and the population layer. For the media layer, we introduce full-time controller $\psi {\dot{B}}_{1}\left(t\right)$ to modify the transmission rate $\alpha_{1}$, i.e., $\alpha_{1}$ becomes $\alpha_{1}+\psi {\dot{B}}_{1}\left(t\right)$. For the population layer, we introduce the intermittent time controller $\sigma {\dot{B}}_{2}\left(t\right)$ to modify the overall removal rate μ, i.e., μ becomes $\mu +\sigma {\dot{B}}_{2}\left(t\right)$. We obtain a non-periodically intermittent stochastic stabilization system:(5)$$d\Xi =F\left(\Xi \right)dt+G_{1}\left(\Xi \right)dB_{1}\left(t\right)+G_{2}\left(\Xi \right)dB_{2}\left(t\right),$$
where
$$\begin{array}{c} \begin{array}{ccc} \Xi \left(t\right) & = & {\left[X(t),Y(t),Z(t,I(t),S(t),T(t),D(t)\right]}^{T},\\ F\left(\Xi \right) & = & \left[\begin{array}{c} -\alpha_{1}X\left(t\right)S\left(t\right)\\ \alpha_{1}X\left(t\right)S\left(t\right)-\alpha_{2}Y\left(t\right)D\left(t\right)\\ \alpha_{2}Y\left(t\right)D\left(t\right)\\ A-\beta_{1}I\left(t\right)S\left(t\right)-\gamma_{1}I\left(t\right)Y\left(t\right)-\mu I\left(t\right)-\beta_{3}I\left(t\right)D\left(t\right)\\ \beta_{1}I\left(t\right)S\left(t\right)+\gamma_{1}I\left(t\right)Y\left(t\right)-\beta_{2}S\left(t\right)D\left(t\right)-\left(\mu +\eta \right)S\left(t\right)-\gamma_{2}S\left(t\right)Z\left(t\right)\\ \eta S(t)-\mu T(t)\\ \beta_{2}S\left(t\right)D\left(t\right)+\gamma_{2}S\left(t\right)Z\left(t\right)+\beta_{3}I\left(t\right)D\left(t\right)-\mu D\left(t\right) \end{array}\right],\\ G_{1}\left(\Xi \right) & = & {\left[-\psi X(t)S(t),\psi X(t)S(t),0,0,0,0,0\right]}^{T}\\ G_{2}\left(\Xi \right) & = & \left\{\begin{array}{c} \sigma {\left(0,0,0,I(t),S(t),T(t),D(t)\right)}^{T}dB\left(t\right),t\in \left[t_{j},t_{j}+w_{j}\right)\\ 0,t\in [t_{j}+w_{j},t_{j}+1) \end{array}\right\}, \end{array} \end{array}$$
for each *j* ∈ *N*, *ψ* represents the full-time control intensity. *σ* represents the non-periodically intermittent perturb intensity. In the rest of this section, a stochastic stabilization criterion on rumor spreading is established by Ito^ formula, exponential martingale inequality, Borel-Cantelli’s lemma, and the stochastic stabilization theory.

**Theorem 4.** 
*If the aperiodically intermittent perturb intensity σ satisfies, $\sigma^{2}>\frac{2}{\omega }\left(\frac{\beta_{1}A}{\mu }-\mu -\eta -\gamma_{2}N_{1}\right)$, the spreader S(t) almost certainly tends to zero exponentiallyfor all k∈N, which means the rumor will die out with probability one.*


**Proof.** By adding the first three equations of the control system (Equation 2), we obtain
$$d\left(X\left(t\right)+Y\left(t\right)+Z\left(t\right)\right)=0,$$
combining the initial values of the system, we can deduce that
$$X\left(t\right)+Y\left(t\right)+Z\left(t\right)=N_{1}.$$
We use $It\hat{o}$ formula to calculate the logarithm of X(t)
$$d\log X\left(t\right)=\left(-\alpha_{1}S\left(t\right)-\frac{1}{2}\psi^{2}{S\left(t\right)}^{2}\right)dt-\psi S\left(t\right)dB\left(t\right),$$
since $S\left(t\right)>0,\;-\alpha_{1}S\left(t\right)-\frac{1}{2}{\alpha_{1}}^{2}{S\left(t\right)}^{2}<0$, we can conclude that X(t) almost certainly decays to 0 exponentially. Moreover,
$$d\left(X\left(t\right)+Y\left(t\right)\right)=-\alpha_{2}Y\left(t\right)D\left(t\right),$$
applying the same analytical method, it is not difficult to deduce that Y(t) also almost certainly approaches zero exponentially, and Z(t) approaches $N_{1}$. To further demonstrate that S(t) almost certainly approaches zero exponentially, we calculate the logarithm of S(t),
$$d\log S\left(t\right)=\left(\beta_{1}I\left(t\right)-\mu -\eta -\beta_{2}D\left(t\right)-\gamma_{2}Z\left(t\right)+\frac{\gamma_{1}I\left(t\right)Y\left(t\right)}{S\left(t\right)}-\frac{{G_{2}\left(t\right)}^{2}}{2{S(t)}^{2}}\right)dt+\frac{G_{2}\left(t\right)}{S\left(t\right)}dB\left(t\right).$$
By integrating both sides of the inequality, we obtain
$$\begin{array}{ccc} \log S\left(t\right) & \leq & \log\left(S(t_{0})\right)+\int_{t_{0}}^{t}(\beta_{1}I\left(\xi \right)-\mu -\eta -\beta_{2}D\left(\xi \right)-\gamma_{2}Z\left(\xi \right)\\ & & +\frac{\gamma_{1}I\left(\xi \right)Y\left(\xi \right)}{S\left(\xi \right)}-\frac{G_{2}{\left(\xi \right)}^{2}}{2S{\left(\xi \right)}^{2}})d\xi +\int_{t_{0}}^{t}\frac{G_{2}\left(\xi \right)}{S\left(\xi \right)}dB\left(\xi \right). \end{array}$$
Set $K_{1}\left(\xi \right)=\beta_{1}I\left(\xi \right)-\mu -\eta -\beta_{2}D\left(\xi \right)-\gamma_{2}Z\left(\xi \right)+\frac{\gamma_{1}I\left(\xi \right)Y\left(\xi \right)}{S\left(\xi \right)}$. Since $\lim_{t\to \infty }Z\left(t\right)=N_{1},$$\;\lim_{t\to \infty }Y\left(t\right)=0$, and $S\left(t\right)\leq \frac{A}{\mu }$, there are positive constants $T_{1}$, *A*, *B*, *C*, *a*, and *b* such that, if $t\geq T_{1}$, $Z\left(t\right)\geq N_{0}-Ae^{-at}$ and $Y\left(t\right)\leq Be^{-bt}$. Then,
$$\begin{array}{ccc} \int_{t_{0}}^{t}K_{1}\left(\xi \right)d\xi & = & \int_{t_{0}}^{T}K_{1}\left(\xi \right)d\xi +\int_{T}^{t}K_{1}\left(\xi \right)d\xi \\ & \leq & \int_{t_{0}}^{T_{1}}K_{1}\left(\xi \right)d\xi +\int_{T_{1}}^{t}\left(\frac{\beta_{1}A}{\mu }-\mu -\eta -\gamma_{2}\left(N_{1}-Ae^{-a\xi }\right)+Ce^{-b\xi }\right)d\xi \\ & = & \int_{t_{0}}^{T_{1}}K_{1}\left(\xi \right)d\xi +\left(\frac{\beta_{1}A}{\mu }-\mu -\eta -\gamma_{2}N_{1}\right)\left(t-T_{1}\right)+\frac{\gamma_{2}A}{a}\left(e^{-at}-e^{-aT_{1}}\right)\\ & & -\frac{C}{b}\left(e^{-bt}-e^{-bT_{1}}\right)\\ & = & \left(\frac{\beta_{1}A}{\mu }-\mu -\eta -\gamma_{2}N_{1}\right)\left(t-T_{1}\right)+K_{2}\left(t\right) \end{array}$$
where $K_{2}\left(t\right)=\int_{t_{0}}^{T_{1}}K_{1}\left(\xi \right)d\xi +\frac{\gamma_{2}A}{a}\left(e^{-at}-e^{-aT_{1}}\right)-\frac{C}{b}\left(e^{-bt}-e^{-bT_{1}}\right)$.By the exponential martingale inequality and Borel-Cantelli’s lemma (see Mao [24] Theorem 7.4 on P44 and Lemma 2.4 on P7), an integer $q_{0}=q_{0}\left(\omega \right)$ exists for almost all ω∈Ω and any positive number *T*, ε, and if $q_{0}>q_{0}\left(\omega \right)$, the continuous martingale
$$\int_{t_{0}}^{t}\frac{G_{2}\left(t\right)}{S\left(t\right)}dB\left(t\right)\leq \frac{2}{\varepsilon }\log\left(q\right)+\frac{\varepsilon }{2}\int_{t_{0}}^{t}\frac{G_{2}{\left(t\right)}^{2}}{S{\left(t\right)}^{2}}dt$$
holds for all 0≤t≤T. This yields that
$$\begin{array}{ccc} logS\left(t\right) & \leq & \log\left(S(t_{0})\right)+\left(\frac{\beta_{1}A}{\mu }-\mu -\eta -\gamma_{2}N_{1}\right)\left(t-T_{1}\right)\\ & & +K_{2}\left(t\right)-\int_{t_{0}}^{t}\left(\frac{1}{2}-\frac{\varepsilon }{2}\right)\frac{G_{2}{\left(\xi \right)}^{2}}{S{\left(\xi \right)}^{2}}d\xi +\frac{2}{\varepsilon }\log\left(q\right). \end{array}$$Since $G_{2}\left(t\right)$ is a piecewise function, we need to consider two distinct cases based on the range of *t* in order to calculate the integral mentioned above. It is evident that a positive integer n exists such that:Case 1. If $t_{n}\leq t\leq t_{n}+\omega_{n}$ and $t_{n}>q_{0}$,
$$\begin{array}{ccc} \log S\left(t\right) & \leq & \log\left(S(t_{0})\right)+\left(\frac{\beta_{1}A}{\mu }-\mu -\eta -\gamma_{2}N_{1}\right)\left(t-T_{1}\right)+K_{2}\left(t\right)\\ & & -\int_{t_{0}}^{t_{0}+\omega_{0}}\frac{\left(1-\varepsilon \right)G_{2}{\left(\xi \right)}^{2}}{2S{\left(\xi \right)}^{2}}d\xi -\int_{t_{0}+\omega_{0}}^{t_{1}}\frac{\left(1-\varepsilon \right)G_{2}{\left(\xi \right)}^{2}}{2S{\left(\xi \right)}^{2}}d\xi \\ & & \cdots -\int_{t+\omega_{n}}^{t_{n}}\frac{\left(1-\varepsilon \right)G_{2}{\left(\xi \right)}^{2}}{2S{\left(\xi \right)}^{2}}d\xi +\frac{2}{\varepsilon }\log\left(q\right)\\ & = & \log\left(S(t_{0})\right)+\left(\frac{\beta_{1}A}{\mu }-\mu -\eta -\gamma_{2}N_{1}\right)\left(t-T_{1}\right)+K_{2}\left(t\right)-\frac{1-\varepsilon }{2}\sum_{i=0}^{n}\omega_{i}\sigma^{2}+\frac{2}{\varepsilon }\log\left(q\right). \end{array}$$Case 2. If $t_{n}+\omega_{n}\leq t\leq t_{n+1}$ and $t_{n}>q_{0}$,
$$\begin{array}{ccc} \log S\left(t\right) & \leq & \log\left(S(t_{0})\right)+\left(\frac{\beta_{1}A}{\mu }-\mu -\eta -\gamma_{2}N_{1}\right)\left(t-T_{1}\right)+K_{2}\left(t\right)\\ & & -\int_{t_{0}}^{t_{0}+\omega_{0}}\frac{\left(1-\varepsilon \right)G_{2}{\left(\xi \right)}^{2}}{2S{\left(\xi \right)}^{2}}d\xi -\int_{t_{0}+\omega_{0}}^{t_{1}}\frac{\left(1-\varepsilon \right)G_{2}{\left(\xi \right)}^{2}}{2S{\left(\xi \right)}^{2}}d\xi \\ & & \cdots -\int_{t_{n}+\omega_{n}}^{t_{n+1}}\frac{\left(1-\varepsilon \right)G_{2}{\left(\xi \right)}^{2}}{2S{\left(\xi \right)}^{2}}d\xi +\frac{2}{\varepsilon }\log\left(q\right)\\ & = & \log\left(S(t_{0})\right)+\left(\frac{\beta_{1}A}{\mu }-\mu -\eta -\gamma_{2}N_{1}\right)\left(t-T_{1}\right)+K_{2}\left(t\right)-\frac{1-\varepsilon }{2}\sum_{i=0}^{n}\omega_{i}\sigma^{2}+\frac{2}{\varepsilon }\log\left(q\right). \end{array}$$
Combined with the above two cases, $logS\left(t\right)$ satisfies
$$\log S\left(t\right)\leq \log S\left(t_{0}\right)+\left(\frac{\beta_{1}A}{\mu }-\mu -\eta -\gamma_{2}N_{1}\right)\left(t-T_{1}\right)+K_{2}\left(t\right)-\frac{1-\varepsilon }{2}\sum_{i=0}^{n}\omega_{i}\sigma^{2}+\frac{2}{\varepsilon }\log\left(q\right)$$
for all $t_{n}\leq t\leq t_{n+1}$, ω∈Ω. Then,
$$\begin{array}{ccc} \frac{1}{t}\log S\left(t\right) & \leq & \frac{1}{t}\log\left(t_{0}\right)+\frac{1}{t}\left(\frac{\beta_{1}A}{\mu }-\mu -\eta -\gamma_{2}N_{1}\right)\left(t-T_{1}\right)\\ & & +\frac{1}{t}K_{2}\left(t\right)-\frac{1-\varepsilon }{2t}\sum_{i=0}^{n}\omega_{i}\sigma^{2}+\frac{2}{\varepsilon t}\log\left(q\right). \end{array}$$
Since ε is arbitrary, setting ε→0, the above inequation satisfies
$$\frac{1}{t}\underset{t\to \infty }{\lim\sup}\log S\left(t\right)\leq -\frac{1}{2}\omega \sigma^{2}+\frac{\beta_{1}A}{\mu }-\mu -\eta -\gamma_{2}N_{1},a.s.$$
If $\frac{1}{2}\sigma^{2}>\frac{\beta_{1}A}{\mu }-\mu -\eta -\gamma_{2}N_{0}$, it yields
$$\frac{1}{t}\underset{t\to \infty }{\lim\sup}\log S\left(t\right)\leq -\frac{1}{2}\omega \sigma^{2}+\frac{\beta_{1}A}{\mu }-\mu -\eta -\gamma_{2}N_{1},a.s.$$
Then, we conclude if $\frac{1}{2}\omega \sigma^{2}>\frac{\beta_{1}A}{\mu }-\mu -\eta -\gamma_{2}N_{1}$, $\lim_{t\to \infty }S\left(t\right)=0,a.s.$ The proof is completed. □

## 5. Numerical Examples

In this section, we conduct numerical simulations to illustrate the validity of the theoretical results.

### 5.1. Parameter Analysis

We analyzed the effect of the relevant system parameters on the criteria of the rumor-free equilibrium, using the partial rank correlation coefficients (PRCCs) method. In order not to lose generality, we set the input parameters to be uniformly distributed (a maximum value 120% of the baseline value; a minimum value 80% of the baseline value) and performed 1000 PRCCs on all parameters, as shown in Figure 2. Figure 2 shows rumor propagation is found to be strongly positively correlated with $\beta_{1}$ and strongly negatively correlated with μ. Our PRCCs results show that reducing the probability of rumor propagation and introducing a rumor detection mechanism can reduce the basic reproduction number.

### 5.2. The Process of Spreading Rumors

In order to verify the stability of the different equilibrium points, we run numerical simulations using the system parameters in Table 2, and the simulations are based on model (Equation 1).

Firstly, we validated the global asymptotic stability of the rumor-free equilibrium point $E_{f}$, as shown in Figure 3. By setting the system parameters according to the first row of Table 2, the results for $E_{f1}$ are depicted in Figure 3a,b. It is evident that the rumor-free equilibrium point $E_{f1}$ is globally asymptotically stable and converges to (0,0,100,1000,0,0,0). Similarly, by setting the system parameters according to the second row of Table 2, the results for $E_{f2}$ are shown in Figure 3c,d. It is apparent that the rumor-free equilibrium point $E_{f2}$ is globally asymptotically stable and converges to (0,0,100,235,0,0,765). The simulation results align with the theoretical analysis.

Secondly, we verified the global stability of the localized rumor equilibrium point $E_{e}$, as illustrated in Figure 4. By setting the system parameters according to the third row of Table 2, the results for $E_{e}$ are presented in Figure 4a,b. It is evident that the localized rumor equilibrium point $E_{e}$ is globally stable, and $E_{e}$ converges to (0,0,100,333,470,59,138), indicating active rumor propagation.

### 5.3. Intermittent Stochastic Control Strategy

In this section, we consider the effect of intermittent control strategies on rumor propagation. Under the local equilibrium point $E_{f2}$, rumors propagate without any control method as shown in Figure 4b. We introduce a variety of intermittent control strategies for different control times and different control intensities in the above case to verify Theorem 4. To better highlight the relationship between the control time ratio and the control intensity, we set up three different combinations, corresponding to three different control intensities under three different control time ratios, as shown in Figure 5a,e,i.

First, we set the control time ratio to ω=0.2 and simulated the time variation of the size of spreaders S(t) for three different control intensities, as shown in Figure 5a. By calculating $\frac{\beta_{1}A}{\mu }-\mu -\eta -\gamma_{2}N_{1}=0.2685$, according to Theorem 4, the random control intensity σ needs to be greater than 0.5182 in order to satisfy the control condition of inhibiting rumor propagation. So, in Figure 5b σ=0.1 is chosen to indicate that rumors cannot be controlled when the control intensity is insufficient. We then adjust the control intensity to σ=0.3 and 0.5, as shown by Figure 5c,d, and the rumor eventually disappears, satisfying the condition in Theorem 4.

Secondly, we set the control time ratio to ω=0.6, as shown in Figure 5e. By calculating $\frac{\beta_{1}A}{\mu }-\mu -\eta -\gamma_{2}N_{1}=0.0895$, according to Theorem 4, the random control intensity σ needs to be greater than 0.2992 in order to satisfy the control condition of inhibiting rumor propagation. So, in Figure 5f σ=0.1 is chosen to indicate that rumors cannot be controlled when the control intensity is insufficient. We then adjust the control intensity to σ=0.3 and 0.5, as shown by Figure 5g,h, and the rumor eventually disappears, satisfying the condition in Theorem 4. Obviously, the greater the intensity of the adjustment control, the better the control effect achieved.

Lastly, we set the control time ratio to ω=0.8, as shown in Figure 5i. The simulation steps are the same as above, and as shown in Figure 5j–l, the results show the effectiveness of intermittent stochastic control strategy.

The overall picture of the study allows us to conclude that the use of random control strategies can be effective in curbing rumor propagation. This suggests that by increasing the intensity and duration of control, we are able to achieve better control results.

## 6. Conclusions

In the process of rumor propagation, the differences between media channels and the individuals involved greatly influence the effectiveness of propagation. Traditional single-layer network models no longer adequately describe the current multi-channel rumor propagation environment. In this study, considering that rumors can be transmitted not only from rumor spreaders but also websites and other media with rumor information, we have established a dual-layer rumor propagation model. Furthermore, we have taken into account the discernment ability of individuals and proposed a rumor detection mechanism. Based on our newly established model, we have analyzed the existence and stability of different equilibrium points and obtained a threshold for discerning the occurrence of rumor propagation. This threshold effectively reflects the promotion of rumor propagation by media outlets and the inhibition of rumor propagation by rumor detectors. To better and more flexibly control rumor propagation, we have designed a class of non-periodic intermittent controllers driven by white noise to supress rumor propagation. The effectiveness of the stochastic stabilization method has been theoretically proven, and specific expressions for the control intensity have been provided. Finally, we have conducted simulation experiments to validate the accuracy of the aforementioned theoretical results.

This article describes the coupling effect between media and the crowd in rumor propagation through the establishment of a dual-layer model. There are still many aspects of the coupling between the media layer and the crowd layer that deserve further investigation, such as the quantitative impact of the media on the crowd’s input rate and rumor propagator conversion rate. These areas also form the basis for our future research directions. Additionally, we will employ more rumor control methods such as pulse control and stochastic optimal control. We will consider using reinforcement learning algorithms for automatic rumor control, such as the reinforcement learning algorithm.

## Figures and Tables

**Figure 1 entropy-25-01192-f001:**
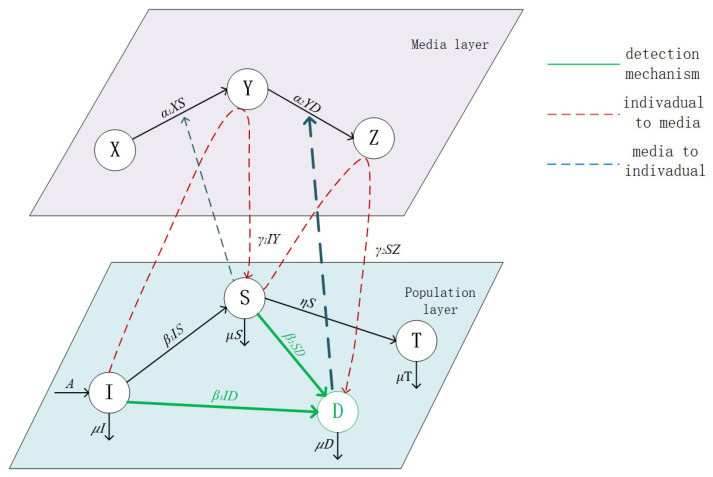
Rumor propagation structure.

**Figure 2 entropy-25-01192-f002:**
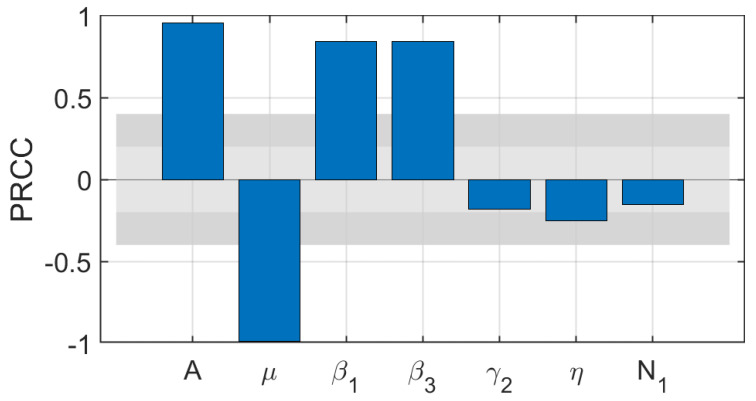
Sensitivity analysis by PRCCs.

**Figure 3 entropy-25-01192-f003:**
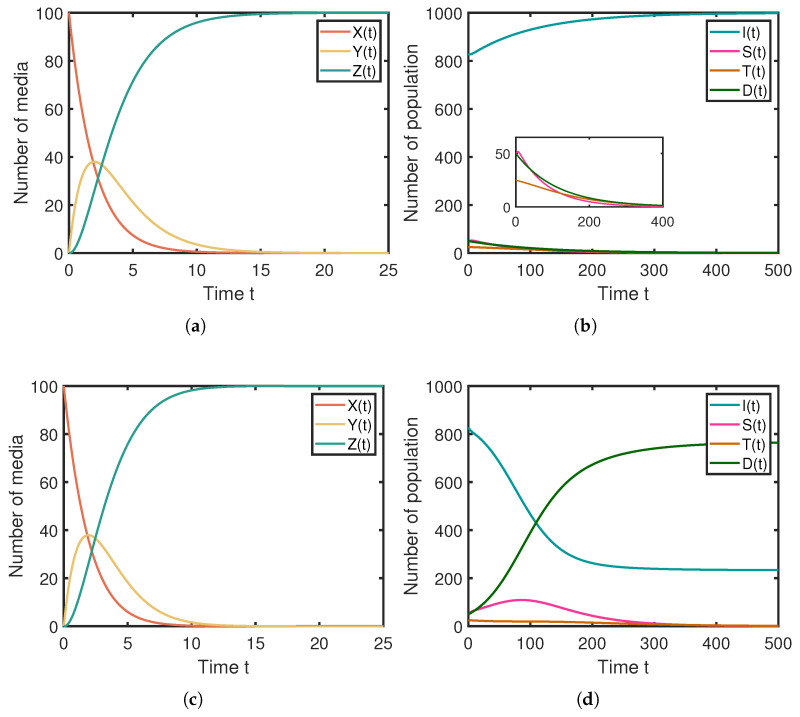
Numerical simulation results for the rumor-free equilibrium point $E_{f}$, where (**a**,**b**) are numerical simulations of $E_{f1}$, (**c**,**d**) are numerical simulations of $E_{f2}$. (**a**) The stablity of rumor-free equilibrium $E_{f1}$ on the media layer. (**b**) The stablity of rumor-free equilibrium $E_{f1}$ on the population layer. (**c**) The stablity of rumor-free equilibrium $E_{f2}$ on the media layer. (**d**) The stablity of rumor-free equilibrium $E_{f2}$ on the population layer.

**Figure 4 entropy-25-01192-f004:**
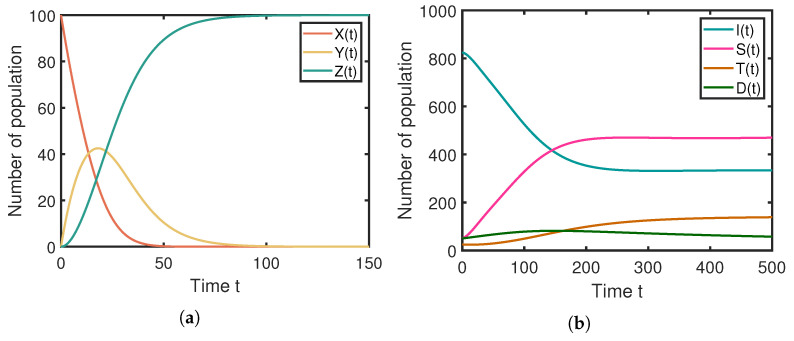
Numerical simulation results for the local rumor prevalence equilibrium point $E_{e}$. (**a**) The stablity of rumor-free equilibrium $E_{e}$ on the media layer. (**b**) The stablity of rumor-free equilibrium $E_{e}$ on the population layer.

**Figure 5 entropy-25-01192-f005:**
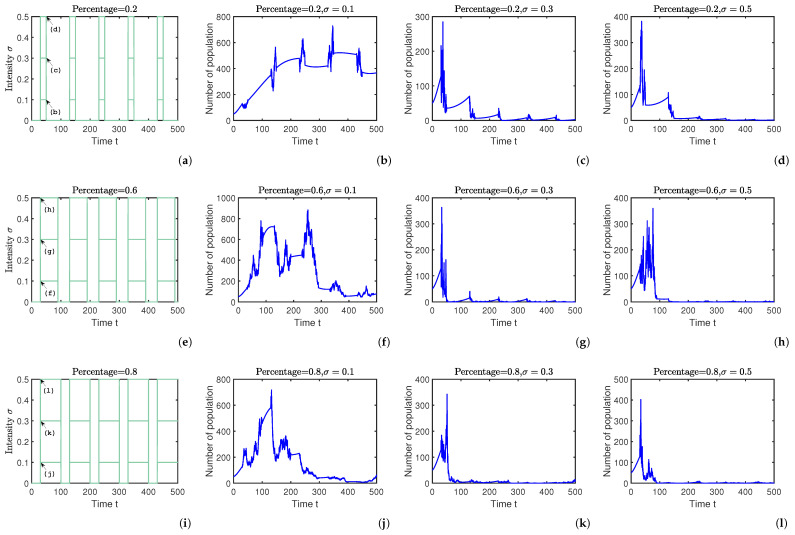
A comparison of intermittent control strategies with three different control times: (**a**) with a control ratio of 20%; (**e**) with a control ratio of 60%; and (**i**) with a control ratio of 80%, all at the local equilibrium point $E_{f2}$. (**b**–**d**) have control intensities of 0.1, 0.3, and 0.5, in that order, at a control time ratio of 0.2. (**f**–**h**) have control intensities of 0.1, 0.3, and 0.5, in that order, at a control time ratio of 0.6. (**j**–**l**) have control intensities of 0.1, 0.3, and 0.5.

**Table 1 entropy-25-01192-t001:** Parameters.

**Symbol**	**Description**
$N_{1}\left(t\right)$	The total number of media in the media layer.
$N_{2}\left(t\right)$	The total number of people in the population layer.
X(t)	Number of susceptible message media at time *t*.
Y(t)	Number of infected message media at time *t*.
Z(t)	Number of immune message media at time *t*.
I(t)	Number of ignoramuses at time *t*.
S(t)	Number of spreaders at time *t*.
T(t)	Number of stiflers at time *t*.
D(t)	Number of debunkers at time *t*.
*A*	Probability of new joiners interested in rumors per unit time *t*.
μ	Probability of moving out of each category in unit time *t*.
$\alpha_{1}$	Probability that X(t) is converted to Y(t) under the influence of spreaders.
$\alpha_{2}$	Probability that Y(t) is converted to Z(t) under the influence of debunkers.
$\beta_{1}$	Probability that I(t) is converted to S(t).
$\beta_{2}$	Probability that S(t) is converted to D(t).
$\beta_{3}$	Probability that I(t) is converted to D(t).
$\gamma_{1}$	Probability that I(t) is converted to S(t) under the influence of infected message media.
$\gamma_{2}$	Probability that S(t) is converted to D(t) under the influence of immune message media.
η	Probability that S(t) is converted to T(t).

**Table 2 entropy-25-01192-t002:** Parameter values.

Parameter	$\alpha_{1}$	$\alpha_{2}$	$\beta_{1}$	$\beta_{2}$	$\beta_{3}$	$\gamma_{1}$	$\gamma_{2}$	η	*A*	μ
values	0.01	0.01	0.000001	0.0000025	0.000001	0.00004	0.0000015	0.003	10	0.01
values	0.01	0.01	0.000032	0.000012	0.000043	0.000055	0.000013	0.0017	10	0.01
values	0.0011	0.0011	0.00004	0.0000025	0.00002	0.00004	0.0000015	0.003	10	0.01

## Data Availability

No data were used to support this study.

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
