# Peer review of "Stochastic Stabilization of Dual-Layer Rumor Propagation Model with Multiple Channels and Rumor-Detection Mechanism"

_entropy, 2023, doi:10.3390/e25081192_

Round 1
Reviewer 1 Report
Recent empirical studies in network science have revealed that real-world interactions often involve multiple channels, extending beyond the traditional framework of a single channel. To address this, multilayer networks have garnered significant attention in recent years for their ability to capture and describe these complex interaction patterns. In this paper, the authors investigate the propagation of rumors across multiple channels and propose a dual-layer rumor propagation model that considers variations in individual cognition. Their findings highlight the influence of the number of media outlets and rumor debunkers on the threshold for rumor disappearance or persistence. Additionally, the authors introduce innovative non-periodic intermittent noise stabilization methods to effectively suppress the spread of rumors.
This work provides a few insights into rumor propagation process in multilayer network. The results are inspirational. Here I have a few comments for the authors to improve their paper.
1.It appears that the integral expression of K_1 in page 10 is missing the \gamma_1 in the inequality on the right-hand side involving the C*e^(-b*\xi). Please double-check for this omission. Besides, the plus sign in front of (gamma_2*A)/a in the next line should be revised to a minus sign.
2.There seems to be an issue with the inequality involving logS(t) (page 10): The \epsilon in the integral term on the right side of the inequality might should be corrected to \epsilon/2? Please review and verify it carefully.
3.Ref.40 is missing.
4.The subfigures (a), (e), and (i) in Fig. 5 should be mentioned in the manuscript.
There are some typos (mistakes), such as: (line 172) "the dyanamic behaviour ..." should be rvised to "the dynamic behaviour ...", (line 173) "persisitence of ruomr spreader" should be revised to "persistence of ruomr spreader", (line 230) "\^Ito formula" should be revised to "It\^o formula", (line 230) "(0, 0, 100, 765, 0, 0, 235)" should be revised to "(0, 0, 100, 235, 0, 0, 765)", etc.
Reviewer 2 Report
1. It's not clear why another rumor propagation model is needed. The authors should identify the problem that cannot be solved using existing models. For example, may be existing models be not enough to eÑ…plain a certain empirical observation.
The motivation for the paper now looks like: There are many models already, so why don't we make one more? It's not a good motivation.
2. Sometimes the text is written so that it is clear only for a narrow circle of those who work with this specific kind of models. For example, what kind of control is "blocking of rumor spreaders" in line 57? What kind of sociological mechanism lies behind this idea?
3. It follows from the bottom formula on page 4 that I(0)+ S(0) + T(0) + D(0) <=A/mju. However the paper never specifies this. It only says that I(0) > 0, S(0) > 0,T(0) > 0, D(0) > 0. I think, it should be explicitly written that the total number of individuals at t=0 is less or equal A/mju.
4. I have noticed one typo: "initial valuas" between lines 146 and 147.
English is OK. Just double-check for unnoticed typos.
Reviewer 3 Report
The authors created a two-layer model to study the spread of rumors, considering information from rumor spreaders, websites, and other media. They also considered individuals' ability to discern rumors, proposed a mechanism for detecting rumors, analyzed different equilibrium points, and determined a threshold indicating when rumor propagation occurs.
The paper is generally well-written; however, a few specific issues need to be addressed.
Methodology
1. The authors should look at the Ignorant–Spreader–Stifler–Checker (ISSC) model (Piqueira et al., 2020), which is the same as their population layer (ISTD)
2. In your model Figure 1 – the D are just a node in input and no output, meaning they are passive nodes that don’t distribute rumors. However, they don’t distribute the truth either. In reality, you have nodes type (D) who actively spread truth and not just don’t spread rumors. I think a model with a separate flow of truth and a process depicts the occlusion between the flow of rumors Vs. The truth will better illustrate your model. The authors should refer to this suggestion or rename their nodes.
3. The authors should address these questions:
· In which cases (and probabilities) X becomes Z without going through Y?
· In which cases (and probabilities) X becomes Z without going through Y?
· In Figure 1 and in the calculations - Why η - the Probability that S(t) is converted to T(t), is not affected by the transform of Y to Z ? (BTW, in Figure 1, ηS is written twice. No need).
· Since I “read” Or “was involved” in X, Why A is the only coming flow to I. How come the volume of X doesn’t affect A in some manner?
Diagrams
It would be beneficial if the authors could present an alternative flow diagram with clearer nodes and edges. Representing the described reality as a data flow diagram (with lanes) or a Markov chain (with probabilities) could help clarify the entire flow of items.
The Model
Although the authors' model is dynamic, it appears somewhat static. In the described chain, people's roles may change over time based on the volume of data (rumors) they are exposed to. A new rule is suggested to be incorporated into the model, allowing nodes to change roles with a suitable probability.
References:
Please refer to the work of:
Piqueira, J. R., Zilbovicius, M., & Batistela, C. M. (2020). Daley–Kendal models in fake-news scenario. Physica A: Statistical Mechanics and its Applications, 548, 123406.
The English is fine.
Reviewer 4 Report
The authors present a model for rumor spreading. They use an ODE framework from the epidemiology that is standard for the field. The model setup and analysis are appropriate and seem fine. The paper can be accepted as is.
Author Response
We appreciate your precious time in reviewing our paper.

Round 2
Reviewer 1 Report
In the revised manuscript, the author has addressed my concerns competently. I would like to recommend it for publication in Entropy.
Reviewer 2 Report
Some comments were made to the original version of the paper. Since necessary improvements have been made, I recommend to accept the revised manuscript.
Reviewer 3 Report
The manuscript has been sufficiently improved to warrant publication in Entropy.